# Comparison of SARS-CoV-2-Specific Antibodies in Human Milk after mRNA-Based COVID-19 Vaccination and Infection

**DOI:** 10.3390/vaccines9121475

**Published:** 2021-12-14

**Authors:** Hannah G. Juncker, Sien J. Mulleners, Marit J. van Gils, Tom P. L. Bijl, Christianne J. M. de Groot, Dasja Pajkrt, Aniko Korosi, Johannes B. van Goudoever, Britt J. van Keulen

**Affiliations:** 1Department of Pediatrics, Emma Children’s Hospital, Amsterdam Reproduction & Development Research Institute, Amsterdam UMC, 1081 HV Amsterdam, The Netherlands; h.juncker@amsterdamumc.nl (H.G.J.); s.j.mulleners@amsterdamumc.nl (S.J.M.); d.pajkrt@amsterdamumc.nl (D.P.); b.j.vankeulen@amsterdamumc.nl (B.J.v.K.); 2Center for Neuroscience, Swammerdam Institute for Life Sciences, University of Amsterdam, 1098 XH Amsterdam, The Netherlands; A.Korosi@uva.nl; 3Department of Medical Microbiology and Infection Prevention, Amsterdam Infection and Immunity Institute, University of Amsterdam, Amsterdam UMC, 1105 AZ Amsterdam, The Netherlands; m.j.vangils@amsterdamumc.nl (M.J.v.G.); t.p.bijl@amsterdamumc.nl (T.P.L.B.); 4Department of Obstetrics and Gynaecology, Amsterdam Reproduction & Development Research Institute, Vrije Universiteit, Amsterdam UMC, 1081 HV Amsterdam, The Netherlands; cj.degroot@amsterdamumc.nl

**Keywords:** breastmilk, immunization (vaccination), SARS-CoV-2, immunoglobulin A, BNT162b2 mRNA COVID-19 vaccine, COVID-19

## Abstract

SARS-CoV-2-specific antibodies are secreted into human milk of infected or vaccinated lactating women and might provide protection to the breastfed infant against COVID-19. Differences in antibody response after these types of exposure are unknown. In this longitudinal cohort study, we compared the antibody response in human milk following SARS-CoV-2 vaccination or infection. We analyzed 448 human milk samples of 28 lactating women vaccinated with the SARS-CoV-2 vaccine BNT162b2 as well as 82 human milk samples of 18 lactating women with a prior SARS-CoV-2 infection. The levels of SARS-CoV-2-specific IgA in human milk were determined over a period of 70 days both after vaccination and infection. The amount of SARS-CoV-2-specific IgA in human milk was similar after SARS-CoV-2 vaccination and infection. After infection, the variability in IgA levels was higher than after vaccination. Two participants with detectable IgA prior to vaccination were analyzed separately and showed higher IgA levels following vaccination compared to both groups. In conclusion, breastfed infants of mothers who have been vaccinated with the BNT162b2 vaccine receive human milk with similar amounts of SARS-CoV-2-specific antibodies compared to infants of previously infected mothers.

## 1. Introduction

The clinical presentation of severe acute respiratory syndrome coronavirus (SARS-CoV-2) in infected children seems to be mild; however, severe coronavirus disease 2019 (COVID-19) has been reported in infants including cases of mechanical ventilation, liver and cardiac function impairment, and even death [1,2,3,4]. The immune system of infants is still developing in the first half year of life, making them more vulnerable to infections [5]. Indeed, based on a recent analysis of 576 SARS-CoV-2-infected children, infants seem to be more vulnerable to severe disease relative to older children [6]. They might be partially protected via human milk, as human milk contains several immune-enhancing factors, including antibodies, oligosaccharides, nucleic acids, and cytokines [7]. The WHO encourages women to continue breastfeeding even during COVID-19 or after vaccination against SARS-CoV-2, because the benefits of breastfeeding outweigh the potential risks [8].

Immunoglobulin A (IgA) comprises most of the antibodies in human milk, and this antibody class plays a major part in mucosal immunity [9], protecting surfaces such as the respiratory epithelium [10]. Previously infected mothers secrete SARS-CoV-2-specific IgA into their milk, and these antibodies are capable of neutralizing the virus [11,12]. The ability to neutralize SARS-CoV-2 is directly associated with levels of IgA against the receptor-binding domain of the viral surface spike (S) protein [13]. Therefore, these neutralizing human milk antibodies may play an important role in protecting the breastfed infant against COVID-19.

Additionally, a SARS-CoV-2-specific antibody response in human milk is observed after vaccination with an mRNA-based SARS-CoV-2 vaccine in lactating women [11,14]. Therefore, vaccination may be especially important for lactating women because it may not only protect the mother from COVID-19 but also her infant through passive transmission of SARS-CoV-2-specific IgA in human milk. It is unknown how antibody levels following vaccination relate to the antibody response observed after an infection; therefore, the aim of this study is to compare antibody levels in human milk after vaccination with an mRNA-based vaccine with antibody levels after SARS-CoV-2 infection.

## 2. Materials and Methods

### 2.1. Study Design and Participants

In this longitudinal cohort study, we aimed to include lactating women who received the mRNA-based SARS-CoV-2 vaccine BNT162b2 and who had a Polymerase Chain Reaction (PCR)-proven SARS-CoV-2 infection. Participants who collected less than three milk samples in the 70 days post-infection were excluded from analysis.

### 2.2. Data Collection

Participants who received the vaccination collected human milk samples according to a set schedule. For each vaccinated participant, 17 human milk samples were collected over a period of 70 days: one sample before the first vaccination and one sample on days 3, 5, 7, 9, 11, and 13, and the last sample between days 15 and 17 after the first vaccination. This schedule was the same for the second vaccination, which was approximately three weeks after the first dose. The last sample was collected approximately 70 days after the first vaccination date. Participants with a PCR-confirmed infection provided a milk sample prior to the start of symptoms, if available, and were instructed to subsequently collect a milk sample every two weeks for at least 70 days. All participants were requested to empty one breast in the morning prior to feeding their child, mix the milk, and store the samples in a freezer until the study appointment.

The characteristics of lactating women and their infants were obtained through a questionnaire that was sent to all participants. Moreover, we requested information on vaccine side effects or COVID-19-related symptoms that the women had experienced.

### 2.3. Data Analysis

After collection, human milk samples were stored at −80 °C at the Amsterdam UMC (VUmc location). An enzyme-linked immunosorbent assay (ELISA) with the SARS-CoV-2 S protein was used to assess the SARS-CoV-2-specific IgA antibodies in human milk as described previously [12]. Briefly, soluble perfusion-stabilized S protein of SARS-CoV-2 was generated and immobilized overnight on a 96-well plate (Greinier) using 0.1 M NaHCO_3_ followed by a one-hour blocking step with 1% casein PBS (Thermo Scientific, Waltham, MA, USA). The human milk samples were diluted at 1:10 in 1% casein PBS (Thermo Scientific, Waltham, MA, USA) and incubated on the S protein-coated 96-well plates for two hours to allow them to bind to the target protein. Finally, a 1:5000 diluted horseradish peroxidase (HRP)-labeled goat anti-human IgA (Biolegend, San Diego, CA, USA) in 1% casein PBS was used to detect specific IgA in the human milk samples. After one hour of incubation, 3,3′,5,5′-tetramethylbenzidine (TMB) was used for the read-out at 450 nm. For the determination of the cutoff value, a relative operating characteristic curve analysis was performed for human milk samples using pre-pandemic negative samples and PCR-proven positive samples. The milk samples were considered positive at an optical density of 450 nm (OD450 nm) and cutoff value of 0.502. With this cutoff value, the sensitivity was 67.9% (95% CI: 61.0–74.1%) for IgA in human milk with a specificity of 99.0% (95% CI: 94.7–100.0%). This is comparable with the sensitivity of IgA in human milk from previous work by Lebrão et al. [15].

### 2.4. Statistical Methods

Descriptive statistics of baseline characteristics were calculated in IBM SPSS Statistics for macOS version 27. Factors that may influence human milk composition were compared between the study groups and included maternal age, body mass index, lactation stage, chronic disease, parity, mode of delivery, vaccination history, ethnicity, birthweight of the infant, gestational age, infant sex, and exclusive breastfeeding.

To compare the antibody responses between both groups, all data collected from the infected group were categorized in 10-day intervals so that each time point consisted of data of at least five human milk samples. The days of positive PCR testing and first vaccine dose were time point 0. To assess the total longitudinal antibody response from time point 0 to 70 days after exposure, we determined the area under the curve with respect to ground (AUCg), with respect to increase (AUCi), and the area under the curve with respect to the cutoff (AUC_cutoff_) as described by Pruessner et al. [16]. The mean total AUC with standard error of the mean (SEM) and degrees of freedom (df) for the vaccinated and infected group were compared using an unpaired *t*-test. AUCs were compared and expressed as mean difference with 95% confidence interval (95% CI). GraphPad Prism version 9.1.2 for macOS was used to display the dynamics of IgA antibodies in human milk over time and to determine and compare the AUCs for both groups [17].

## 3. Results

Human milk samples were collected from 46 lactating women, of whom 28 received the mRNA-based COVID-19 vaccine BNT162b2 and 18 had a prior PCR-confirmed SARS-CoV-2 infection (Figure 1).

### 3.1. Baseline Characteristics

The age of the participants ranged from 25 to 41 years, and the majority had been breastfeeding for 6 to 12 months. The baseline characteristics did not differ between vaccinated or infected mothers (Table 1). Two of the participants who were vaccinated showed milk conversion at baseline and were therefore handled as a separate group.

### 3.2. Side Effects following Vaccination

Detailed information regarding adverse events following vaccination with BNT162b2 are reported in Table 2. One or more side effects were reported by 57% of the participants after the first dose and by 81% after the second dose. Local pain and muscle aches were the most reported side effects after the first vaccination, while fatigue and fever were the most frequently reported after the second dose.

### 3.3. COVID-19 Symptoms

Data on infection-related symptoms was obtained from twelve out of eighteen infected participants (Table 3). Most of the participants had mild symptoms, of which the most commonly reported were headache (75%), fatigue (75%), and sudden loss of smell and/or taste (63%). One participant was asymptomatic, and one was admitted to the hospital as a consequence of COVID-19.

### 3.4. Human Milk IgA Levels following Vaccination

After vaccination, a biphasic IgA response was observed, reaching a peak at two weeks after the first dose (Figure 2). Human milk IgA started to rise again five days after the second dose and reached a second peak after one week. After the first dose was administered, milk conversion was observed in 19 participants (73%) after 15 days on average (which ranged from 7 to 27 days). After the second dose, 25 participants (96%) reached milk conversion. One participant showed no milk conversion after receiving both doses. This participant was similar in baseline characteristics compared to the other vaccinated mothers and did experience a headache and fever after vaccination. After 70 days, 7 out of 18 participants (39%) still had detectable SARS-CoV-2-specific IgA. 

### 3.5. Human Milk IgA Levels following SARS-CoV-2 Infection

IgA levels were measured in 82 human milk samples from 18 previously SARS-CoV-2-infected participants. These levels after infection are displayed in Figure 2. Participants showed milk conversion after a median of 15 days. After 35 days, a peak in IgA was observed, after which, IgA gradually declined to undetectable levels at 56 days. Seventy days after PCR confirmation of SARS-CoV2 infection, 33% of the human milk samples contained detectable IgA levels. In total, six (33%) previously infected participants showed no milk conversion during our study. These participants were similar in age, ethnicity, BMI, chronic disease, and lactation stage compared with the rest of the infection group. Overall, participants without milk conversion experienced similar symptoms to the participants with detectable antibodies in their milk. The only difference was that fewer reported the sudden loss of smell.

### 3.6. IgA Titers over Time: Vaccination Versus Infection

Over the study period, the AUC_G_ was 37.6 ± 6.4 after vaccination and 38.2 ± 5.6 after infection (Table 4). The mean difference in AUC_G_ was 0.6 (95% CI, −29.14 to 30.3; *p* > 0.9) and was not statistically significant. The mean AUC_I_ over 70 days was 19.8 ± 6.4 after vaccination and 19.7 ± 5.6 after infection. The mean difference was 0.13 (95% CI, −29.9 to 29.6; *p* = 0.99) and was not statistically significant. The AUC above the cutoff (AUC_cutoff_) was 7.5 (95% CI, 1.3 to 14.9) after vaccination and 5.7 (95% CI, 0.0 to 15.4) after infection. The mean difference was −1.8 (95% CI, −19.6 to 16.10; *p* = 0.85) and was not statistically significant. Table 4 shows that the AUCs for the first 35 days after vaccination or infection did not differ between the study groups.

### 3.7. IgA Variability

SARS-CoV-2-specific IgA levels were more variable among infected participants compared to vaccinated participants: AUC variances were higher (F = 7.1; *p* < 0.0001). Interindividual variability in the infection group was also reflected in the wider 95% CI of mean IgA levels (Figure 2, filled area between error bars indicates 95% CI of mean IgA, *p* < 0.01). 

### 3.8. Vaccination of Previously Infected Mothers

Two participants initially included in the vaccination group showed milk conversion before the administration of the first vaccine dose. After vaccination, both women showed an accelerated and higher immune response compared to women without detectable antibodies at baseline. Moreover, SARS-CoV-2-specific IgA antibodies remained detectable during the 70 days of follow-up.

SARS-CoV-2-specific IgA levels in the human milk of the two participants vaccinated after a previous PCR-confirmed SARS-CoV-2 infection were higher than those in the infection group and the vaccination group: mean AUC_G_ of 71.1 ± 8.3 (*p* < 0.05) and mean AUC_cutoff_ of 36 ± 8.3 (*p* < 0.05). The IgA levels of these two participants were more stable over time (Figure 2), with a mean AUC_I_ of 12.3 ± 8.3 (*p* = 0.86). Moreover, only one peak was observed in the IgA level after vaccination of the previously infected lactating women.

## 4. Discussion

SARS-CoV-2-specific IgA levels in human milk were similar after vaccination with the BNT162b2 vaccine and following a SARS-CoV-2 infection over a period of seventy days. On an individual level, a higher variability in human milk antibody responses was observed after infection rather than vaccination. 

Our findings are in line with previous research on serum where IgA levels were similar following infection or vaccination [18]. In human milk, the IgA response was studied separately following infection or vaccination [19,20,21], but only a few studies compared both exposure types longitudinally. In one study, higher IgA levels were observed in human milk of previously infected participants at day 12 versus those in vaccinated participants 47 days after receiving the first dose [22]. In contrast, Low et al. [23] reported significantly higher IgA titers in human milk following vaccination versus human milk samples collected five months after infection and those collected one month after vaccination. Earlier research in serum found that the SARS-CoV-2-specific IgA antibody response is correlated with the time since exposure and seems to play an especially important role during early exposure as a first line of defense [10,11]. Therefore, the SARS-CoV-2-specific antibody levels of SARS-CoV-2-vaccinated or infected mothers at different time points after exposure should not be compared. 

Interestingly, we found significantly higher levels of milk IgA after vaccination in previously infected participants. This amplifying effect was already demonstrated by several studies for serum antibodies where one dose of the BNT162b2 vaccine elicited significantly higher antibody levels in the serum of previously infected participants [24,25,26]. 

Consistent with previous reports [21], we found high variability in human milk SARS-CoV-2-specific IgA levels in previously infected participants, with some of them having no milk conversion over the entire study period. In contrast, almost all vaccinated participants showed milk conversion. The antibody response during a SARS-CoV-2 infection is known to vary with viral load and symptom severity [27]. Less variability was observed following vaccination, as expected for vaccines designed and tested to elicit a certain immune response. 

This study has several limitations that should be mentioned. Although we collected a large number of samples, the number of participants in each group was limited. Second, the collection schedule was different between groups with fewer sample collection days in the infected group, which could partially explain the higher variability in this group. Moreover, the baseline for the infected participants was set at the day of the positive PCR test result. This could have led to a pattern shift in IgA titers over time because on average, symptoms start five days after exposure to SARS-CoV-2 [28]. Another limitation is that no conclusions can be drawn from our research regarding the protective properties of the elicited antibodies, as we did not determine the neutralization capacity of IgA in human milk, the infection rate in infants, or reinfection in our participants. However, previous studies have demonstrated the neutralization capacity of SARS-CoV-2-specific IgA in human milk following vaccination [13,22] and infection [20,21]. Moreover, human milk antibodies have proven reductive effects on infectious disease in nursing infants [29,30].

## 5. Conclusions

Over a period of 70 days, the human milk of lactating women vaccinated with the mRNA-vaccine BNT162b2 contained similar SARS-CoV-2-specific IgA levels as lactating women with a SARS-CoV-2 infection. These human milk antibodies might protect the breastfed infant, which is warranted because infants might be at risk of developing severe COVID-19. Collectively, the beneficial effects of breastfeeding, the small risk of transmission during a maternal infection [31], and the unlikely possibility that mRNA from the vaccine will be transferred to the infant [23], advocate for continued breastfeeding after both SARS-CoV-2 vaccination or infection. 

## Figures and Tables

**Figure 1 vaccines-09-01475-f001:**
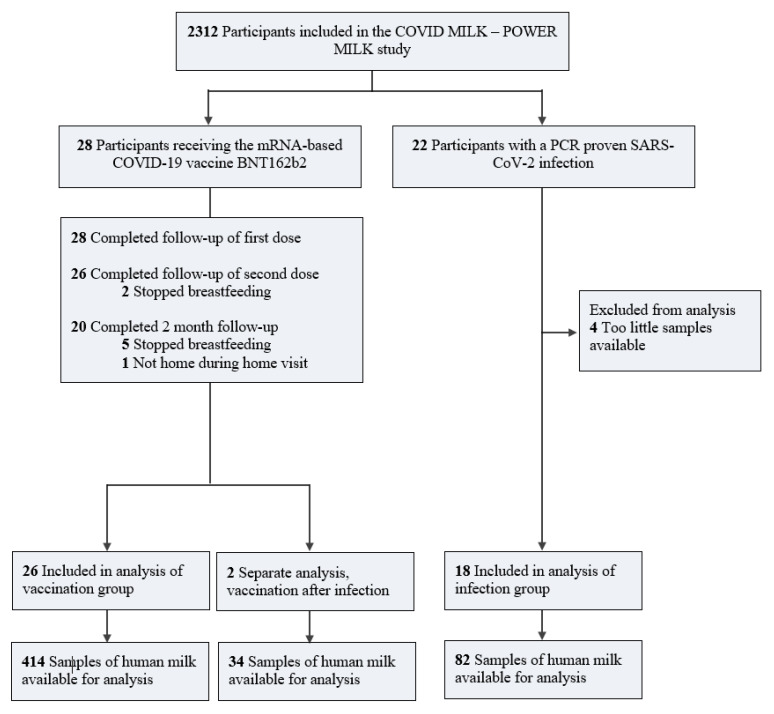
Flow diagram of lactating women participating in this study. BNT162b2 indicates a COVID-19 vaccine; mRNA, messenger ribonucleic acid; PCR, polymerase chain reaction; SARS-CoV-2, severe acute respiratory syndrome coronavirus 2.

**Figure 2 vaccines-09-01475-f002:**
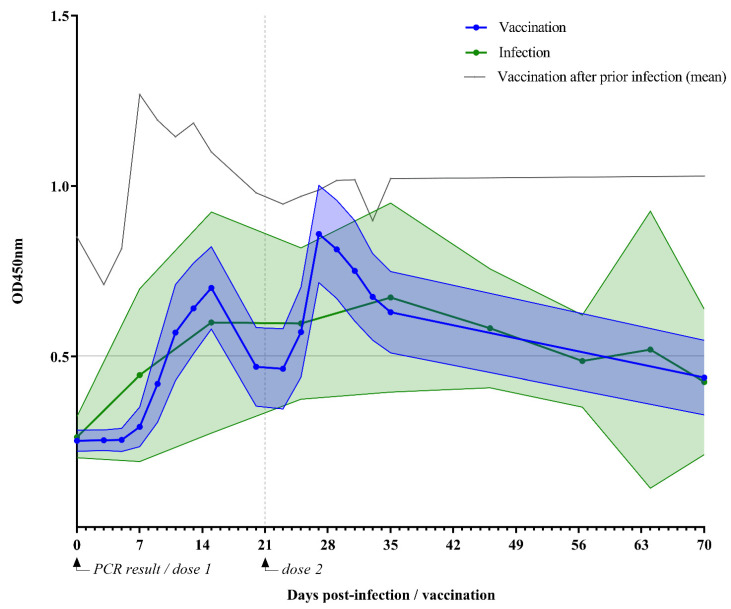
SARS-CoV-2-specific IgA levels following vaccination or infection. OD450 nm indicates optical density 450 nm. This figure shows specific IgA levels in human milk of lactating women over 70 days after vaccination with the mRNA-based COVID-19 vaccine BNT162b2 (*n* = 26) or a naturally acquired SARS-CoV-2 infection (*n* = 18). The solid dotted lines indicate mean IgA level in human milk following vaccination (blue) or infection (green); filled area between error lines indicates 95% confidence interval. The grey solid line indicates the mean IgA level in human milk following vaccination after prior infection (*n* = 2). The horizontal grey line at 0.502 represents the cutoff value for the detection limit of IgA in human milk.

**Table 1 vaccines-09-01475-t001:** Characteristics of participants at first visit.

	BNT162b2 Vaccination		SARS-CoV-2 Infection
	No Milk Conversion (*n* = 26)	Milk Conversion (*n* = 2) ^1^	(*n* = 18)
Maternal features			
Age, median (IQR), y	33.5 (30.8–35)	35.5	31 (28.5–36.5)
BMI, median (IQR), kg/m2	24.0 (21.0–26.6)	25.8	25.8 (21.6–25.2)
Chronic disease, No. (%)	3 (11.5) ^2^	0 (0)	0 (0)
Parity primiparous, No. (%)	12 (46.2)	1 (50)	9 (50)
Age postpartum, median (IQR), months	7.0 (4.8–9.0)	9	6.0 (4.5–8.5)
Delivery mode (vaginal), No. (%)	19 (73.1)	1 (50)	12 (66.7)
Lactational stage, No. (%)			
<6 months	8 (30.8)	1 (50)	5 (27.8)
6–12 months	16 (61.5)	-	11 (61.1)
>12 months	2 (7.7)	1 (50)	2 (11.1)
Vaccination history, No. (%)			
Child-immunization ^3^	26 (100)	2 (100)	16 (88.9)
Whooping cough vaccine during pregnancy	25 (96.2)	2 (100)	15 (83.3)
Other	14 (53.8) ^4^	1 (50)	12 (66.7) ^4^
Race/ethnicity, No. (%)			
Europe	25 (96.2)	2 (100)	14 (77.8)
Oceania	1 (3.8)	-	2 (11.1)
Africa	-	-	1 (5.6)
South America	-	-	1 (5.6)
Infant features			
Birthweight, median (range), grams	3388 (3126–3710)	3495	3475 (3278–3848)
Gestational age, mean (SD), weeks	39.1 (1.7)	41	39.9 (1.5)
Sex of infant (female), No. (%)	16 (61.5)	1 (50)	7 (38.9)
Exclusive breastfeeding, No. (%)	12 (46.2)	1 (50)	7 (38.9)

Abbreviations: IQR, interquartile range; BMI, body mass index; SD, standard deviation; ^1^ All values for two previously infected vaccinated women are presented in this table as mean; ^2^ Diabetes Mellitus type 1, epilepsy, hypothyroidism; ^3^ According to the Dutch National Immunization Program; ^4^ other vaccinations specified: hepatitis B and A, yellow fever, rabies, typhoid fever, influenza A, and cholera.

**Table 2 vaccines-09-01475-t002:** Side effects after administration of the mRNA-based COVID-19 vaccine BNT162b2.

Side Effects ^1^	No. (%)
After first dose	
Local pain/swelling	6 (28.6)
Muscle aches	5 (23.8)
Headache	4 (19)
Fever	1 (4.8)
Fatigue	2 (9.5)
Other	2 (9.5) ^2^
After second dose	
Fatigue	7 (33.3)
Local pain/swelling	6 (28.6)
Fever	5 (23.8)
Headache	5 (23.8)
Muscle aches	4 (19)
Other	9 (42.9) ^3^
No side effects at both doses	1 (3.6)

^1^ Numbers in this table were computed with data obtained from 21 participants. Of the total 28 vaccinated participants, adverse events were unknown for 7. ^2^ Other side effects specified at dose 1: insomnia. ^3^ Other side effects specified at dose 2: hypothermia, general malaise, mastitis, joint pain, diarrhea, and chills.

**Table 3 vaccines-09-01475-t003:** Symptoms related to PCR-diagnosed SARS-CoV-2 infection.

Symptoms	No. (%) ^1^
Headache	12 (75)
Fatigue	12 (75)
Sudden loss of smell and/or taste	10 (62.5)
Cold symptoms	9 (56.3)
Sore throat	8 (50)
Fever	6 (37.5)
Dry cough	6 (37.5)
Photophobic	2 (12.5)
Abdominal pain	1 (6.3)
Nausea/vomiting	1 (6.3)
Loss of appetite	1 (6.3)
Hospital admission ^2^	1 (6.3)
Asymptomatic	1 (6.3)

Abbreviations: PCR, reverse transcription polymerase chain reaction. Data in this table represent symptoms of 16 participants. None of the participants experienced diarrhea, shortness of breath, or tachypnoea. ^1^ Symptoms related to PCR-confirmed SARS-CoV-2 infection were unknown for 2 of the 18 participants in the proven infection group. ^2^ No intensive care unit admission.

**Table 4 vaccines-09-01475-t004:** Area under IgA titer curves over one month and 70 days.

70 Days	Vaccination, Uninfected	Vaccination, Previously Infected	SARS-CoV-2 Infection, Unvaccinated
df	397	17	73
AUC_G_	37.6 ± 6.4 ^a^	71.1 ± 8.3 ^b^	38.2 ± 5.6 ^a^
AUC_cutoff_	7.5 ± 3.8 ^a^	36 ± 8.3 ^b^	5.7 ± 4.9 ^a^
AUC_I_	19.8 ± 6.4 ^a^	12.3 ± 8.3 ^a^	19.7 ± 5.6 ^a^
One month			
df	380	16	44
AUC_G_	18.9 ± 1.8 ^a^	35.2 ± 1.1 ^b^	19 ± 4.3 ^a^
AUC_cutoff_	5.6 ± 1.6 ^a^	17.7 ± 1.1 ^b^	3.7 ± 4 ^a^
AUC_I_	10 ± 1.8 ^a^	6.2 ± 1 ^a^	9.8 ± 4.3 ^a^

Values are in mean ± SEM. Abbreviations: SEM, standard error of the mean; df, degrees of freedom; AUC_G_, area under the curve with respect to ground; AUC_cutoff_, to IgA cutoff value set at 0.502; AUC_I_, to increase. ^a, b^ Different superscript letters next to AUC values indicate a significant difference. Statistical significance (*p* < 0.05) was determined using a one-way ANOVA test.

## Data Availability

Data will be available to researchers who provide a methodologically correct proposal to achieve the aims in the approved proposal. Proposals should be directed to the corresponding author to gain access to the data. Data requestors need to sign an access agreement.

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
