# Peer review of "Comparison of SARS-CoV-2-Specific Antibodies in Human Milk after mRNA-Based COVID-19 Vaccination and Infection"

_vaccines, 2021, doi:10.3390/vaccines9121475_

Round 1

Reviewer 1 Report

This small-scale study looks for IgA antibody persistence in milk after COVID19 infection vs vaccination. The study is well-executed and presented nicely. As the authors acknowledged, the cohort size is smaller but this study is able to fulfill its aim of comparing the trend of antibody presence/persistence in milk of infected vs vaccinated women over time. Overall, a good study. My only suggestion will be to discuss the no milk converters from both vaccinated and infected cohorts with respect to their sociodemographic and other characteristics. 

Author Response

This small-scale study looks for IgA antibody persistence in milk after COVID19 infection vs vaccination. The study is well-executed and presented nicely. As the authors acknowledged, the cohort size is smaller but this study is able to fulfill its aim of comparing the trend of antibody presence/persistence in milk of infected vs vaccinated women over time. Overall, a good study. My only suggestion will be to discuss the no milk converters from both vaccinated and infected cohorts with respect to their sociodemographic and other characteristics. 

Reply: Thank you for reviewing our article and your positive evaluation of our work. Thank you for your suggestion to discuss the characteristics of the participants without milk conversion. We described the characteristics and symptoms of the participants without milk conversion in relation to the rest of their group in the result sections:

  • Lines 170 – 172 “This participant was similar in baseline characteristics compared to the other vaccinated mothers and did experience a headache and fever after vaccination.”
  • Lines 190-194: “These participants were similar in age, ethnicity, BMI, chronic diseases or lactation stage compared to the rest of the infection group. Overall, participants without milk conversion experienced similar symptoms as the participants with detectable antibodies in their milk. They only reported fewer sudden loss of smell.

Reviewer 2 Report

This paper reports very interesting data on a still poorly investigated topic: the levels of anti-SARS-CoV2 IgA in the milk of mothers exposed to SARS-CoV-2 infection or to RNA-based vaccination. Despite the limited size of the study population, the paper presents original data on the kinetics of IgA production and secretion in the milk. I would suggest to the authors to include in the methods section also the timing of first-dose and second-dose vaccination. Furthermore, I would suggest to analyse if there is any correlation between the levels of SARS-CoV-2 RNA at diagnosis and the amount of IgA in the milk or if severity of symptoms after vaccination correlated with a higher IgA production. Lastly, do the authors have any data on IgG production in serum of mothers, was there any correlation between IgA levels in milk and IgG in serum?

Overall, I support the publication of this paper after these minor revisions.

Author Response

This paper reports very interesting data on a still poorly investigated topic: the levels of anti-SARS-CoV2 IgA in the milk of mothers exposed to SARS-CoV-2 infection or to RNA-based vaccination. Despite the limited size of the study population, the paper presents original data on the kinetics of IgA production and secretion in the milk.

Thank you for reviewing our article and your constructive suggestions. Please find below our response and the adjustments we have made in response to these suggestions.

I would suggest to the authors to include in the methods section also the timing of first-dose and second-dose vaccination.

Reply: We have added in the method section that the second dose was approximately three weeks after the first dose (line 75).

Furthermore, I would suggest to analyse if there is any correlation between the levels of SARS-CoV-2 RNA at diagnosis and the amount of IgA in the milk or if severity of symptoms after vaccination correlated with a higher IgA production.

Reply: Thank you for your suggestion. Although it is a very interesting suggestion, we do not have information on SARS-CoV-2 RNA levels at diagnosis so we are not able to add this information in the paper. Also the other suggestion, aiming at quantifying the severity of symptoms to IgA expression is a very interesting one. However, this study was not designed as such and consequently our sample size is not sufficient to draw conclusions on this question. Moreover, a validated questionnaire to classify severity of symptoms is currently not available which makes it difficult to investigate this relationship. However, we do describe the symptoms of participants without milkconversion in relation to the rest of their group that did show antibodies.

  • Lines 170 – 172 “This participant was similar in baseline characteristics compared to the other vaccinated mothers and did experience a headache and fever after vaccination.”
  • Lines 190-194: “These participants were similar in age, ethnicity, BMI, chronic diseases or lactation stage compared to the rest of the infection group. Overall, participants without milk conversion experienced similar symptoms as the participants with detectable antibodies in their milk. They only reported fewer sudden loss of smell.

Lastly, do the authors have any data on IgG production in serum of mothers, was there any correlation between IgA levels in milk and IgG in serum?

Reply: Thank you for this interesting suggestion. We did consider to look at the correlation between IgA and IgG. However as this is not completely in the scope of this specific paper and as we also did not measure IgG in serum as frequently as we measured IgA in human milk (only 1 to 4 times for each participant), we decided not to include serum IgG levels. We expect that these antibodies do not strongly correlate with each other due to their different functions in the immune response. IgA plays a key role in the initial immune response as the first line of defense against the virus and wanes after a relatively short period, while IgG is predominantly important in the secondary immune response and is maintained for months. 

Overall, I support the publication of this paper after these minor revisions.

Reply: thank you, we hope that you agree with our responses above and the adjustments we have made according your suggestions. Your suggestions are very helpful for next studies.

Reviewer 3 Report

This is an interesting and well-designed study. A few minor comments for improved clarity. 

  1. The sampling schedules do not match with what is shown in Fig 2, including day 17 after 1st dose and all time points after 2nd dose. Please clearly state what time points the samples were collected, and make sure they match with those shown in Fig 2.
  2. Fig 2: what does the dot line at OD of 0.5 indicate? If this is the detection limit, clearly state in the figure legend. Add the number of subjects per group in the figure legend. 

Author Response

This is an interesting and well-designed study. A few minor comments for improved clarity.

Thank you for your suggestions to improve our article. Please find below our responses to your suggestions.

  1. The sampling schedules do not match with what is shown in Fig 2, including day 17 after 1st dose and all time points after 2nd dose. Please clearly state what time points the samples were collected, and make sure they match with those shown in Fig 2.

Reply: thank you for your comment. We did collect the last sample after each vaccination between day 15 and day 17 and not on both days. We agree that it is not fully clear at this moment in the method section and therefore we adjusted this sentence into: “one sample before the first vaccination and one sample on days 3, 5, 7, 9, 11, 13, and the last sample between days 15 and 17 after the first vaccination” (line 74). We also adjusted Figure 2 to make it more clear. We hope that these clarifications are sufficient.

  1. Fig 2: what does the dot line at OD of 0.5 indicate? If this is the detection limit, clearly state in the figure legend. Add the number of subjects per group in the figure legend.

Reply: thank you for this useful reply. We adjusted the figure legend according to your suggestions. The figure legend is now: “SARS-CoV-2-specific IgA levels following vaccination or infection. OD450nm , indicates optical density 450 nanometer. This figure shows specific IgA levels in human milk of lactating women over 70 days after vaccination with the mRNA-based COVID-19 vaccine BNT162b2 (n=26) or a naturally-acquired SARS-CoV-2 infection (n=18). The solid dot-ted lines indicate mean IgA level in human milk following vaccination (blue) or infection (green); filled area between error lines indicates 95% confidence interval. The grey solid line indicates the mean IgA level in human milk following vaccination after prior infection (n=2). The horizontal grey line at 0.502 represents the cutoff value for the detection limit of IgA in human milk.”